# Impact of Physical Activity on Cognitive Functions: A New Field for Research and Management of Cystic Fibrosis

**DOI:** 10.3390/diagnostics10070489

**Published:** 2020-07-18

**Authors:** Valentina Elce, Alessandro Del Pizzo, Ersilia Nigro, Giulia Frisso, Lucia Martiniello, Aurora Daniele, Ausilia Elce

**Affiliations:** 1MoMiLab, IMT School for Advanced Studies, Piazza San Francesco 19, 55100 Lucca, Italy; valentina.elce@imtlucca.it; 2Dipartimento di Fisica, University of Pisa, Largo Bruno Pontecorvo, 3, 56127 Pisa, Italy; alessandro.delpizzo@df.unipi.it; 3Dipartimento di Scienze e Tecnologie Ambientali, Biologiche, Farmaceutiche, Università della Campania Luigi Vanvitelli, Via Vivaldi, 81110 Caserta, Italy; nigro@ceinge.unina.it (E.N.); aurora.daniele@unicampania.it (A.D.); 4CEINGE Biotecnologie Avanzate SCarl, Via Gaetano Salvatore 486, 80145 Napoli, Italy; gfrisso@unina.it; 5Dipartimento di Medicina Molecolare e Biotecnologie Mediche, Università degli Studi di Napoli Federico II, 80131 Napoli, Italy; 6Dipartimento di Scienze Umanistiche, Università Telematica Pegaso, Via Porzio, Centro Direzionale, isola F2, 80143 Napoli, Italy; lucia.martiniello@unipegaso.it

**Keywords:** cystic fibrosis, brain, physical activity

## Abstract

Cystic Fibrosis (CF) is a genetic disease inherited by an autosomal recessive mechanism and characterized by a progressive and severe multi-organ failure. Mutations in Cystic Fibrosis Conductance Regulator (CFTR) protein cause duct obstructions from dense mucus secretions and chronic inflammation related to organ damage. The progression of the disease is characterized by a decline of lung function associated with metabolic disorders and malnutrition, musculoskeletal disorders and thoracic deformities, leading to a progressive decrement of the individual’s quality of life. The World Health Organization (WHO) qualifies Physical Activity (PA) as a structured activity produced by skeletal muscles’ movements that requires energy consumption. In the last decade, the number of studies on PA increased considerably, including those investigating the effects of exercise on cognitive and brain health and mental performance. PA is recommended in CF management guidelines, since it improves clinic outcomes, such as peripheral neuropathy, oxygen uptake peak, bone health, glycemic control and respiratory functions. Several studies regarding the positive effects of exercise in patients with Cystic Fibrosis were carried out, but the link between the effects of exercise and cognitive and brain health in CF remains unclear. Animal models showed that exercise might improve learning and memory through structural changes of brain architecture, and such a causal relationship can also be described in humans. Indeed, both morphological and environmental factors seem to be involved in exercise-induced neural plasticity. An increase of gray matter volume in specific areas is detectable as a consequence of regular training in humans. Neurobiological processes associated with brain function improvements include biochemical modifications, such as neuromodulator or neurohormone release, brain-derived neurotrophic factor (BDNF) production and synaptic activity changes. From a functional point of view, PA also seems to be an environmental factor enhancing cognitive abilities, such as executive functions, memory and processing speed. This review describes the current state of research regarding the impacts of physical activity and exercise on cognitive functions, introducing a possible novel field of research for optimizing the management of Cystic Fibrosis.

## 1. Introduction

Cystic Fibrosis (CF) is a congenital systemic disease inherited as an autosomal recessive disorder, with manifestation since birth, characterized by chronic and progressive symptoms [1]. The disease is caused by mutations that occur in Cystic Fibrosis Conductance Regulator (*CFTR*) gene, encoding a protein largely expressed in epithelial cells, which has the role of regulating the chloride efflux across the cellular membrane. Mutations in *CFTR* gene lead to protein degradation, protein misfunctions or protein mislocalization [2,3]. Consequently, disturbances in the flow of sodium and chloride ions through cellular membrane occur, resulting in the production of dense secretions, which lead to a progressive obstruction of epithelial ducts of multiple organs and systems—the respiratory trait, exocrine glands, intestine, etc. Chronic illness in CF is closely associated to progressive respiratory system impairment due to the accumulation of high-density mucus, which determines the establishment of chronic inflammation and specific bacterial colonization [4,5]. The severity of the disease has an important impact on CF patients’ quality of life, since it involves not only health, but leads to serious psychological and social impairments. Indeed, depression and anxiety symptoms in adults and adolescents with CF have been recently described [6]. The development of different therapies, i.e., pharmacologic therapy, physio-respiratory therapy, physical exercise, nutritional intervention, etc., have significantly improved both the life quality and life expectancy of CF patients. However, progression of the disease is unavoidable, leading to an increase of therapies and hospitalizations required during life. Frequent disease exacerbations lead to a diffuse sedentariness among CF patients, who show a marked decrease in exercise tolerance, partly related to impaired muscle function and malnutrition, in addition to lung diseases [7]. Due to the complexity of each patient’s health status, exercise interventions are conceived in a multidisciplinary approach, including medical supervision, nutritional and psychosocial management, self-management strategies and mucus clearance techniques. Regular exercise is highly recommended in the care guidelines for CF patients [8], not only because it contributes to mobilize mucus and to enhance pulmonary functions, but also because it can produce health improvements such as reduction of anxiety and depression episodes and an increase of sleep quality [9]. However, a recent Cochrane review highlighted that a great portion of the studies on CF and PE lack strong evidence and that high-quality randomized control trials are required in order to understand the general aspects and the role of physical exercise in CF [10]. The benefits obtained from including training in the CF patients care program are related to adherence to the training, but also the type, duration and frequency of proposed activity (aerobic or anaerobic training) can directly influence the outcome. No negative side effects were identified in order to discourage regular exercise in CF patients, but further studies are required to assess the benefits and the types of activity that should be suggested to CF patients to improve clinic outcomes, such as peripheral neuropathy; oxygen uptake peak; bone health; glycemic control; and impacts of the type and frequency of training on of Forced Expiratory Volume in one-second percentage (FEV1%) [10]. In addition, a series of neurocognitive disfunctions were observed both in children and adult CF [11,12].

Physical Activity (PA) represents any body movement produced by skeletal muscles, and thus, requires energy consumption [13]. In this sense, any motor behavior, including daily and leisure activities [14] falls into the definition of PA. Several studies described the association between PA and wellbeing, cognition and academic achievement, both from biological and psychological points of view. These studies suggest that PA positively affects not only physical and mental health, but also brain health and cognition, i.e., the set of mental processes allowing the acquisition, storage, manipulation and retrieval of information at different levels. As described by Mandolesi and colleagues in 2018 [15], it seems that Physical Exercise (PE), which is a planned, structured and repetitive PA (e.g., aerobic and anaerobic activity) characterized by precise frequency, duration and intensity, is an “enhancer environmental factor promoting neuroplasticity.” Neuroplasticity plays a crucial role for learning, memory, brain development and brain repair [16], since it allows the nervous system to change structurally and functionally in response to environmental demands, physiological changes, injuries and experiences occurring across lifespan. The brain may react to these events through the production or the induction of signaling pathways of neuromodulators or neurohormones, thereby changing the efficacy of synaptic connections, removing existing connections or creating new ones. Therefore, structural and functional changes in the brain are mutually related: any morphological alterations at cellular and molecular levels correspond to modifications in neuronal efficiency and functionality. Conversely, functional changes induced by environmental factors (such as physical activity) may alter the brain anatomy.

The very first evidence regarding the relationship between PA, neural plasticity and cognitive and brain health comes from animal models. Several studies showed that an enriched captive environment, offering increased opportunities for learning, socializing activities and PA (e.g., climbing structures, hiding places and foraging tasks), provided benefits for animals’ cognitive abilities [17]. Studies on rodents, indeed, found that animals housed in impoverished cages displayed significantly different neurobiology than rodents housed in enriched cages [18,19,20]. Moreover, rodents housed in an enriched environment with a running wheel and sustained cognitive stimulation showed enhanced learning and memory consolidation. These findings in animal studies have raised interesting questions regarding the effect of PA on cognitive and brain health in humans. Indeed, following the hypothesis that environmental enrichment can affect brain functioning by inducing changes at molecular and cellular level, previous studies have tried to assess whether and how different kinds of PA can affect human brain functioning at different ages, both in healthy subjects and in clinical populations.

The aim of this review is to describe the correlation between PA and mental health improvements, by analyzing the positive effects of PA on brain health in general population and the molecular mechanisms responsible for them. In addition, the psychological symptoms and the neurocognitive functions’ impairment found in CF patients will be addressed. The aim of the paper is to highlight the existing knowledge about CF and neurocognitive functions impairments, proposing a starting point to speculate about the possibility of developing further studies on the link existing between the effects of PA and the neurocognitive functions of CF patients. These studies may lead to a general improvement of management of CF patients, especially in CF children.

## 2. Psychological Symptoms and Neuronal Impairment in CF

In recent years, several studies regarding psychological diseases related to CF were carried out. The International Depression/Anxiety Study (TIDES) reported a high prevalence of psychological diseases, such as anxiety and depression, in both adults and adolescents with CF [21]. A successive cross-sectional study by Cronly et al. confirmed that depression and anxiety symptoms are related to a low BMI and a decrease of pulmonary function [22].

Few studies have analyzed the relationship between CF and brain health/cognitive functions and cognitive development in CF patients. The limitations of studies concerning nervous system abnormalities in CF are determined by the scarce possibility to select and study CF patients without confounding factors, such as CF-related complications/pharmacological interventions, i.e., children with CF that do not suffer potential malnutrition and absence of various treatments interfering with cognitive functions.

*CFTR* expression was found not only in epithelial cells, but also in other cellular types—neurons, lymphocytes, etc.; however, its role in these cellular types remains yet unclear [23]. Studies concerning in situ hybridization highlighted *CFTR* expression in human neurons from the central nervous system, including the brain and spinal cord [24,25,26,27,28], and in human neurons within the peripheral neuronal systems, especially in cells of the sympathetic and parasympathetic networks of the enteric system [27]. The task of the enteric system is to connect the digestive and intestinal traits with the central system. The ganglial expression of *CFTR* in the enteric system may partially explain the typical gastrointestinal symptoms in CF (viz. distal intestinal obstruction syndrome, meconium ileus of the newborn, small bowel bacterial overgrowth, rectal prolapse, gastroparesis and gastroesophageal reflux). *CFTR* expression was found in ventricular heart cells, smooth muscles and endothelial cells [29,30]. Moreover, a recent study by Lidintong and colleagues found that CFTR regulates cerebral artery myogenic tone; there was a strong correlation between cerebral artery CFTR protein expression and cerebral perfusion. In the study, the authors observed that CFTR corrector compounds (C18 or lumacaftor) normalize pathological alterations in cerebral artery *CFTR* expression, vascular reactivity and cerebral perfusion of mouse models of heart failure and subarachnoid hemorrhage, without affecting systemic hemodynamic parameters [31]. In these pathologies, massive production of Tumor Necrosis Factor (TNF) occurs [32,33]. Autocrine /paracrine TNF signaling in cerebral arteries stimulates sphingosine-1-phosphate (S1P) production with the consequence of downregulating *CFTR* expression [34]. A decline of neurocognitive functions was described both for surviving patients with heart failure and subarachnoid hemorrhage. There are no studies in the literature regarding cerebral perfusion analysis in CF patients. Some nervous system abnormalities are reported in CF patients and in heterozygous subjects for *CFTR* gene mutations [35,36,37].

Animal models, such as *CFTR^−/−^* pigs, at birth, display intestinal lesions (meconium ileus and microcolon), exocrine pancreatic destruction and gallbladder abnormalities [38], but also features of both axonal and demyelinating neuropathy, found in CF patients [36,37]. Rezinov LR et al., in 2013, observed a significant reduction in nerve conduction velocity and an increased incidence of axonal myelin infoldings in *CFTR^−/−^* pig [38].

Piasecki B. et al., in 2016, by administering the Wisconsin Card Sorting Test (WCST), observed that CF children present abnormalities in executive functions, memory and attention. These dysfunctions are associated with logical, conceptual and abstract thinking deficits; lower cognitive flexibility; and a reduced ability to change the course of reasoning in response to changing circumstances, if compared to healthy subjects and children with inflammation bowel disease [11]. These findings are in accordance with other studies on CF adults [39] and on CF patients suffering from end-stage pulmonary disease awaiting lung transplantation [40,41]. Executive functions are often altered in other respiratory diseases, such as asthma and obstructive apneas, suggesting that in CF patients the most probable key mechanism linked to neurocognitive function alterations is not directly related to CFTR malfunctioning, but it may be the correlated with hypoxia. A similar decline of cognitive functions maybe due to factors related to hypoxia and inflammation [12] and not to anxiety and depression disturbs, as observed in end-stage pulmonary disease patients undergoing transplantation. CF children have a high prevalence of sleep disturbances [42], characterized by poor sleep efficiency and frequent arousal, with a total mean sleep duration lower than seven hours per night. Unlike adults, children have specific sleep duration requirements. Actigraphy and polysomnography studies revealed that CF children tend to spend more time in wakefulness after sleep onset, compared to healthy control subjects [43,44]. It was observed that Forced Expiratory Volume in one-second rate (FEV1%), adopted in CF to predict residue lung function, is closely related to sleep efficiency and latency in CF patients. Patients with a reduced FEV1% show a lower sleep efficiency and sleep duration [45]. In healthy children, inadequate sleep is associated with reduction of life quality, poor academic performance, inattentiveness and behavioral problems [46]. The frequency of these deficits, particularly with regard to attention and executive functioning, is high among children with obstructive sleep apnea (OSA). Dancey and colleagues found an association between poor sleep quality and difficulties concerning serial addition and subtraction tasks in adult CF patients [47]. Children and adults with CF who report poor subjective sleep quality also report reduced quality of life [48], especially among the adolescents, including physical, social and emotional functioning; vitality; and health perception [45]. Poor mood is another consequence of inadequate sleep among CF children [49].

Insufficient sleep has been associated with the presence of elevated serum inflammatory markers, even among healthy subjects [50].

## 3. Physical Activity/Exercise and Cognitive Enhancement in Humans

From a functional point of view, research on middle-aged and older adults reported improvements in cognitive functioning related to PE. More specifically, cross-sectional and epidemiological studies [51,52,53], aimed at identifying lifestyle factors that reduce cognitive decline in aging, evidenced fitness-related improvements in memory, attentional processes and executive functions. In particular, executive functions result crucial in selection, scheduling, coordination and monitoring of complex, goal-directed behavior. Systematic literature reviews have evidenced the beneficial effects of motor activity on executive functions, leading some to hypothesize that PE mostly affects vascularization, neural growth and synaptic connections in the prefrontal cortex, the brain region related by previous studies to executive functioning [54]. As far as memory is concerned, a recent study explored the effects of different types of acute exercise on memory functioning [55] by comparing open-skilled exercise, for which the participant was required to dynamically react to a changing and unpredictable environment (e.g., badminton, racquetball), to a closed-skilled exercise, where instead, the environment was predictable and self-paced (e.g., running, walking). The authors found that closed-skilled exercise prominently affected retrospective memory, i.e., the ability to recall past events, while such a difference was not observed for prospective memory, crucial for performing a planned action and recalling a planned intention. In older populations, physical exercise has been linked to a reduced risk of developing dementia [56,57] and to the decrease in the deterioration of executive functions [58]. Moreover, intervention studies in middle-aged and older adults, wherein researchers tried to assess the chronic effects of fitness programs lasting for weeks or months on cognitive functions, reported significant improvements in executive functions, attention, memory and speed of processing, specifically after endurance aerobic training (e.g., running, swimming, cycling or walking). Results were compared to populations practicing non-endurance training (e.g., light stretching, toning programs) or to sedentary control groups [56]. Even a single exercise session was also found to have an acute effect on cognitive functioning: in these studies, cognitive performances were quantitively measured right before and immediately after a single bout of aerobic or resistance exercise, which lasted from few minutes up to several hours. The authors observed an enhancement of older adults’ executive functioning that was independent of the type or duration of the exercises, with the greatest benefits during the post-exercise stage [59].

PE has also been related to cognitive enhancement, and consequently, to academic achievement in children [54,60]: comparisons between children practicing regular aerobic activity and sedentary children of the same age [61] showed the former’s better performance on verbal, perceptual and arithmetic tests. As reported by Donnelly and colleagues [54], PA programs and physical fitness in children positively affected self-regulation and behavioral inhibition. Their accuracy and speed in cognitive test performance were also increased; specifically, the latter improved after individual short-term bouts of fitness. Interestingly, a recent study [62] compared global and regional gray matter volumes, assessed by magnetic resonance imaging, and academic achievement of metabolically healthy obese/overweight (MHO) children to those of metabolically unhealthy (MUO) peers, also taking into account the effect of cardiorespiratory fitness. Participants were defined as unhealthy according to the presence of altered values for any of four risk factors: triglycerides, glucose, high-density lipoprotein and systolic and/or diastolic blood pressure. Academic achievement was measured through the Spanish version of the Woodcock–Johnson III test. Cardiorespiratory fitness was evaluated with a 20 m shuttle run test, where children were asked to run a 20 m distance following an audio signal, and at each reiteration the speed was incremented. The authors found marginally greater global gray matter volume and significantly greater regional gray matter volume in MHO children as compared to MUO children. In MHO participants, global gray matter volume was in turn positively correlated to academic achievement. However, when the analyses were adjusted for cardiorespiratory fitness, the relationship between gray matter volume and academic achievement was borderline non-significant, suggesting a role of fitness in modulating brain development in different metabolic phenotypes, that, due to the lack of similar studies, needs to be further explored.

From a neurobiological point of view, several findings attested that in healthy adults, PE induced significant structural changes, such as increased gray matter volume in frontal and hippocampal regions [63,64]. PE also seems to promote the release of neurotrophic factors, such as peripheral brain-derived neurotrophic factor (BDNF), the increase of blood flow and the improvement of cerebrovascular health [57,65]. Moreover, studies based on Positron Emission Tomography (PET) reported that higher cardiorespiratory fitness was related to changes in metabolic networks related to cognition in subjects at risk of dementia [66]. Functional magnetic resonance imaging (fMRI) research evidenced more efficient neuronal activity in the prefrontal regions after aerobic exercise in normal aging subjects and in patients affected by mild cognitive impairment (MCI) and Alzheimer’s Disease (AD) [67,68]. Focusing on clinical populations, a recent review [69] evidenced the potential of PE in preventing both AD risk and cognitive decline in affected patients. As stated by Valezuela and colleagues, previous studies reported the beneficial effects of PE on cardiovascular risk factors involved in the pathogenesis of Alzheimer’s disease (e.g., diabetes, reduced vascular flow). Moreover, PE seems to lighten the pathophysiological features of AD, including amyloid-β deposition, by exerting anti-inflammatory effects and improving brain redox status. Similarly, Groot et al. demonstrated that PA interventions are equally beneficial in AD patients and in subjects affected by a form of non-AD dementia. The authors also demonstrated that both aerobic and non-aerobic exercise and both high and low-frequency exercise have a positive effect on cognitive functions [70].

In the light of the framework outlined, the picture is far from clear. Even though there seems to be moderate albeit promising evidence that PA might enhance cognitive abilities, meta-analyses of literature on the relationship between PA and brain functioning in both healthy subjects and clinical populations revealed a huge variability across studies. Moreover, there is little evidence regarding the positive effect of PA across the lifespan, since most studies focused on pre-adolescent children and older adults. According to Erikson and colleagues [17], the lack of consistent results could be linked to several factors, including heterogeneity in study designs, both in type and quality of PA measurements and in tools for quantitatively and qualitatively assessing cognitive functions. With regard to the latter, as described above, previous findings attest to executive functions as the cognitive domain that is most affected by PA. However, the authors [17] suggest that these results may be misleading, since several studies selectively focused on describing the effects of PA on executive functions, which were frequently assessed by administering to the participants traditional cognitive tests primarily conceived for clinical diagnosis of dementia rather than for the evaluation of cognitive development in healthy subjects.

## 4. Neurobiological and Environmental Mechanisms Regarding the Impact of PA/PE on Cognitive Functions

Research does not agree yet on a neurobiological explanation for the effect of PA on executive functions. As reported by Mandolesi and colleagues [57], the cognitive improvements observed in training populations could be led back to the roles of both morphological and environmental factors in neural plasticity (see above). More specifically, the cerebral reserves hypothesis, firstly proposed in 2002 by Yakov Stern regarding a possible relationship between Alzheimer’s Disease (AD) onset and education level, provides interesting hints about the potential of both neuroanatomical characteristics and environment in affecting brain functioning and subjects’ susceptibility to functional impairment in clinical conditions [71]. Stern observed that AD patients’ cognitive decline varied significantly according to their specific education level: in other words, those with higher education levels seemed to display slower cognitive decline, even though the anatomical damages were similar to those of patients with lower education levels. According to Stern, this could be explained by assuming the existence of specific brain reserves (BR) and cognitive reserves (CR) that, based on their capabilities, allow the brain to react differentially to both environmentally demanding and physiological or pathological changes. As described clearly by Mandolesi and colleagues [57], while BR concerns neuroanatomical features at the cellular and molecular level (e.g., brain size, neuronal density and synaptic connectivity), which in a neurodegenerative disease also define the “amount of brain damage that can be sustained before reaching a threshold for clinical expression” CR refers to the functional efficiency of neural circuits, measured through behavioral tasks and related to the environmental stimulations provided across life span. CR may allow coping or compensatory mechanisms after brain damage and may affect the susceptibility to functional impairment. Therefore, following the cerebral reserves hypothesis, physical activity could be interpreted as a crucial environmental factor for the specific enrichment of cognitive reserves, along with high-demand cognitive activities, correct sleep quality and a healthy lifestyle.

Nevertheless, other neurobiological dynamics may underlie such a relationship between physical activity and cognitive and brain health [71]: PA may positively affect sleep behavior, which in turn might enhance cognitive functions. Functional improvements might be related to modification of insulin/glucose signaling induced by PA, or rather, to the release of neurotrophic factors, such as BDNF (see above); or to the increase of blood flow consequent to physical exercise [17]. Moreover, other variables could affect cognitive function without any direct link: for example, a higher educational background, socioeconomic status or a healthy lifestyle. Therefore, based on current literature, it is not possible to precisely describe a causal relationship between physical activity and cognitive improvement yet, and further studies are undoubtedly required to shed light on this connection, excluding the influence of external variables from the analysis.

Finally, PE might also positively affect psychological wellbeing, and therefore, the quality of life. While research on children found that PE is correlated with increased self-efficacy and perceived competence, in youth and adulthood it resulted in enhancing both the self-concept and self-esteem [15]. Moreover, in studies on elderly populations, PE was associated with independence that, in turn, seems to favor social interactions and mental health. PE seems even to modulate mood, personality traits and the development of the self both in clinical populations and in healthy subjects: for example, previous studies reported that, in patients with major depressive disorder, aerobic exercise, with an intensity between 30 and 70% of maximal heart rate, positively affects the treatment by significantly reducing depressive symptoms [72]; similar results were also achieved in populations practicing anaerobic exercise (e.g., yoga or activities with rhythmic abdominal breathing and repetitive movements which do not provide for interpersonal competition). Furthermore, previous studies attested that single bouts of exercise may also be helpful for the treatment of anxiety disorders, regardless of the specific type of activity [15]. However, duration results to be a significant variable in defining the psychological benefits of physical activity: more specifically, longer training programs, lasting several months, are more beneficial in the reduction of anxiety and depression symptoms than shorter programs lasting only few days. According to these findings [8], from a neurobiological point of view, such correlations between PE and psychological improvements may be explained through the modulation of peripheral BDNF, as explained previously.

## 5. Role of Physical Activity in Neurocognitive Functions in CF: State of the Art

The positive effects of PA in patients affected by peripheral pathological conditions from metabolic conditions to lung disorders have been widely described [73,74,75,76]. In diseases with an impairment of lung function (similar to that of CF patients), such as chronic obstructive pulmonary disease (COPD) or asthma, PA acts by reducing the risk of developing depression or anxiety symptoms over time [77,78].

While there is evidence of an association between PA and slower decline in pulmonary function in patients with CF, little evidence is present in the literature about the role of PA in brain functions in CF [79]. Bradley et al. found that patients with moderately severe CF develop hypoxemia and hypercapnia more frequently during exercise and sleep in comparison with healthy subjects with similar respiratory muscle strength and nutritional status [80]. Respiratory muscle weakness and malnutrition are independent of the risk of developing hypoxemia or hypercapnia during exercise or sleep [72].

An important point that is still unclear is the relationship between the global exercise-induced metabolic improvements and neurocognitive functions in CF. In our experience, in a group of fifty-nine adult patients with CF that performed regularly supervised physical exercise for 3 years, we observed that PE had significantly beneficial effects on CF patients, influencing parameters such as FEV1% decline; anthropometric parameters (lower number of cases with altered BMI, waist and arm circumferences); lipid and glucose metabolism; and vitamin D serum levels. All these factors may contribute to cognitive changes that are yet to be analyzed [81]. In another study, we observed that CF patients who performed regular physical exercise for at least three years showed improved inflammation status, via immune-metabolic processes involving adiponectin, leptin and C-reactive protein [82]. In the active group of CF patients, we found that triglyceride levels were significantly lower than the sedentary group. Adiponectin serum levels were significantly lower in active people with respect to sedentary CF patients. On the contrary, leptin levels resulted slightly decreased in sedentary CF patients compared to active patients, although the difference was not significant. Finally, C-reactive protein levels were significantly lower in the active group than in the sedentary group.

No papers regarding the relationships of cognitive functions’ responses to regular physical exercise in CF have been published yet. These themes may represent a new field of research. Interestingly, a recent paper by Schneider EK et al. highlighted the emerging effect of the combination of potentiator ivacaftor and the corrector lumacaftor on central and peripheral nervous systems of CF patients. These two drugs are actually adopted to correct and potentiate the CFTR activity in cases of mutations that lead to a mis-function and/or mislocalization of the protein. Ivacaftor and its metabolites, hydroxymethylivacaftor (iva-M1) and ivacaftorcarboxylate (iva-M6), display a significant affinity to the 5-hydroxytryptamine (5-HT; serotonin) 5-HT2c receptor, β3-adrenergic receptor, δ-opioid receptor and dopamine transporter with radioligand binding assays. Iva-M1 has affinity for the 5-HT2c receptor and the muscarine M3 receptor, while iva-M6 shows affinity for the 5-HT2A receptor. In the same work, the intraperitoneally administration of ivacaftor for 21 days in a mouse model of chronic depression leads to reduced immobility of the mouse and to higher locomotor activity with respect to the control, caused by M3 and β3 receptor allosteric modulation. The treatment of ivacaftor and lumacaftor in CF adults shows improvements in anxiety and depression, but also increased sleep, energy and exercise participation, improved appetite and weight gain [83]. Table 1 reports all neurocognitive alterations described in CF patients in the literature and previously summarized in this review.

## 6. Conclusions

Non-pharmacological therapies, such as physical activity interventions, are an appealing alternative as a complementary treatment for several diseases. In the last few years, more and more studies focused on the specific effects of PA/PE on the health status of the general population ranging from childhood to the older population, partially showing the molecular mechanisms responsible for those effects (Table 1, Figure 1). The evidence regarding the effects of PA on health at every age is becoming more and more solid and is spreading to several fields. The few data available have shown that beneficial effects on neurocognitive functions can be expected in CF patients (Figure 1). Further studies are required to understand the effects of exercise, pharmacologic treatment and mental health on CF, and therefore, it represents a novel, promising and unexplored research field to be investigated, in order to improve the management and quality of life of Cystic Fibrosis patients.

Using this review as a starting point, the authors hope to develop studies that compare the cognitive performances of CF patients practicing PA and those of sedentary CF patients.

## Figures and Tables

**Figure 1 diagnostics-10-00489-f001:**
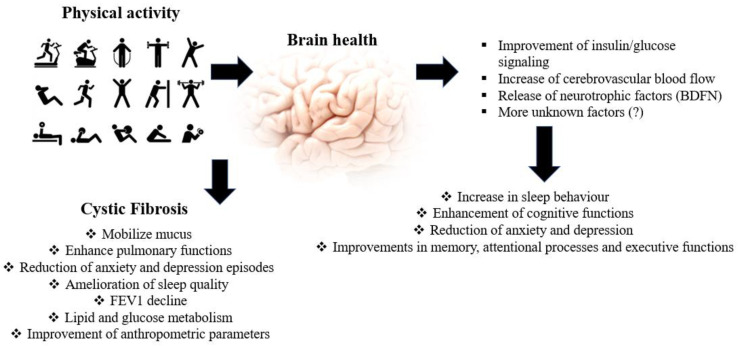
Effect of physical activity on Cystic Fibrosis patients (on the left) and on brain functions (on the right).

**Table 1 diagnostics-10-00489-t001:** Neurocognitive functions’ alterations and psychological diseases in CF subjects.

Neurocognitive Alteration/Psychological Symptoms	Authors
Depression and anxiety symptoms	Cronly J et al., 2018 [22]
Executive functions, memory and attention alterations	Piasecki B et al., 2016 [11]
Logical, conceptual and abstract thinking deficits Lower cognitive flexibility Reduced ability to change the course of reasoning in response to changing circumstances	Maddrey MA et al., 1998 [39] Crews WDJR et al., 2000 [40] Crews WDJR et al., 2003 [41] Hoffman BM et al., 2012 [12]
Sleep disturbances	Bradley S et al., 1999 [80] De Castro-Silva C et al., 2010 [43] Spicuzza L et al., 2012 [44] Button BM et al., 2016 [84] Vandeleur M et al., 2017 [45]
Worsening of anxiety and depression in female adolescent patients on modulators therapy	McKinzie et al., 2017 [85]

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
