# Peer review of "Impact of Physical Activity on Cognitive Functions: A New Field for Research and Management of Cystic Fibrosis"

_diagnostics, 2020, doi:10.3390/diagnostics10070489_

Round 1
Reviewer 1 Report
Summary
The manuscript entitled, “Relationship between brain health, cognitive functions and physical activity in Cystic Fibrosis” provides a contemporary and novel review of literature and top for consideration that has received relatively little attention in people with cystic fibrosis (CF) to date. A general comment that I have regarding this article is that whilst the authors present and highlight a topic that is novel and of real interest, particularly when we consider aspects of CF care such as children studying, more and more adults working now, however it needs a substantial rewrite and refocus in several places. It is not until very late in the article that we really start to discuss cognition in CF and then the links between PA/exercise and cognition and brain health in CF. This focus needs to come through and more of the contemporary knowledge regarding how CF disease alters exercise pathophysiology and discuss potential extension of our knowledge of peripheral oxygenation and vascular function to this topic and aspects such as cerebrovascular function in people with CF.
Summary (Abstract):
I feel that the order of this section needs some work focus on the key messages to be conveyed. Cystic fibrosis should be introduced earlier, as well as the clinical importance of physical activity and fitness in this population.
On Line 31 you outline that one of the primary benefits of PA in people with CF is improving peripheral neuropathy, I disagree with this. There is a wealth of literature now outlining the key functional, clinical and prognostic benefits of physical activity and different types of exercise in people with CF and some of this important, contemporary information needs to come across in your article.
Line 32 – ‘peak oxygen’ does not make sense here. I know that you are referring to peak oxygen uptake (VO2peak), but this needs to be introduced better. Additionally, being physically active is not necessary a sufficient stimulus to increase VO2peak – intensity of exercise training in key in this regard.
Line 36 – ‘the state of the art in this field..’ – this sentence does not make sense in its current written form.
Review
As per comments on your summary (abstract) section, there is a need to restructure the content in this section I would suggest. There is a only a very short paragraph introducing cystic fibrosis which seems to come from nowhere and, as such, it is not immediately clear to the reader why you are now talking about cystic fibrosis. This should come early, mention that this topic has received little attention, but then introduce the wider area of physical activity and cognition and why this should now be considered in people with cystic fibrosis.
You state that the aims are to report the correlation between PA and mental health improvements, however this review is much wider than this. I suggest that the review aims need to be rewritten to better reflect this and the nature of the present article.
Line 96 – ‘a growing number of researches regarding CF..’ poor written English that needs a rewrite
You mention CFTR expression in various tissues that may be mechanistically relevant in the deterioration of cognition in people with CF, but do not include vascular endothelial cells. Given suggested links between cerebral vascular function and cognition, it this not worth consideration? You mention this in your figure but there is no comprehensive discussion.
Line 153 - There was a recent review published by Valenzuela et al. (2020) in Ageing Research Reviews concerning ‘Exercise benefits on Alzheimer’s disease: State-of-the-science’ that the authors may wish to include in the present introduction sections.
A general comment that I have regarding this article is that whilst the authors present and highlight a topic that is novel and of real interest, particularly when we consider aspects of CF care such as children studying, more and more adults working now, however it needs a substantial rewrite and refocus in several places. It is not until very late in the article that we really start to discuss cognition in CF and then the links between PA/exercise and cognition and brain health in CF. This focus needs to come through and more of the contemporary knowledge regarding how CF disease alters exercise pathophysiology and discuss potential extension of our knowledge of peripheral oxygenation and vascular function to this topic and aspects such as cerebrovascular function in people with CF. Really quite important aspects such as ageing within CF have also not been considered or discussed.
Reviewer 2 Report
Summary/Major comment:
With the advent of high effective CFTR modulators (exelecaftor-tezacaftor-ivacaftor), patients with cystic fibrosis (CF) are now expected to have significant increase in length of life. For this reason, research evaluating non-pulmonary health outcomes such as cognitive function is important. In this paper, the authors attempt to review current literature and describe the current state of research examining impact of physical activity on cognitive function and "brain health" in patients with CF. This review of literature does discuss that CFTR is present within neurons and provide plausible mechanism by which CFTR dysfunction/CF may impair brain health. However, there is very limited research describing impact of CF on brain health and even less work to review on impact of physical activity on CF specific cognitive outcomes. For this reason, I feel the title is somewhat misleading. I would suggest title change. As the authors state in line 310 (No papers regarding the relationship between cognitive functions response to regular physical exercise in CF have been published yet), there is not evidence to describe the relationship between physical activity and brain health specifically in cystic fibrosis. I might suggest a title more in line with the current literature such as something like... "Impact of CF on brain health: call for research examining impact of physical activity on cognitive outcomes."
General comments by section:
Introduction
Authors provide general description about relationship between physical activity and brain health and neural plasticity, but provide very little information on reason this relationship is important in cystic fibrosis. I would recommend further discussion as to why in the current state of CF care, understanding this relationship important. As described above, final sentence of introduction - we will define the psychological symptoms and the neurocognitive functions impairment that occur in CF patients and the state of the art about the effects of PA on CF patients, with a specific regard to neurocognitive functions. Highlighted statement is misleading as the authors do not find evidence to review regarding effects of PA on CF patients neurocognitive functions.
Psychological symptoms and neuronal impairment in CF
The section summarizes various reports suggesting that CF impacts various neurologic systems. These findings overall needs conclusive summative links to overarching goal which is to describe impact of CF on cognitive and brain health. For example, the final few sentences describes impact of CF on sleep disturbances but do not describe how sleep disturbances may impact cognitive function.
Role of physical activity on neurocognitive functions in CF: state of art
301-308 Authors state "we have observed that CF patients which perform regular physical exercise for at least three years showed an improved inflammation status, via immune-metabolic processes involving adiponectin, leptin, and C-reactive protein" How are these findings important as related to cognition and brain health? would provide evidence supporting link.
Regarding Table 1.
Would repeat literature search. Their are additional publications that should be added. Example below:
Paper describing worsening anxiety and depression in female patients on modulators.
Journal of cystic fibrosis
Volume 16, Issue 4, July 2017, Pages 525-527
Conclusions:
Although the effects of PA/PE have been studied in many genetic diseases, to our knowledge there are no published studies about inherited diseases. This statement is confusing and needs clarification. Are authors meaning to state no published studies evaluating PA/PE and impact on cognitive function? Additionally, there are numerous studies reporting on exercise and cognitive function in inherited genetic disease such as in Alzheimers disease.
Figure 1.
would change FEV1% decline to reduced rate of FEV1 decline
Other general comments
Throughout the paper there are various language issues or statements that are unclear and frequent revisions will be required. Few examples below:
The development of multiple strategies of cures, i.e. pharmacologic therapy, physio-respiratory therapy, physical exercise, nutritional intervention, etc. have significantly improved the quality and the expectancy of life of CF patients. - These are treatments but do not cure the disease. Do date there is no cure for CF. Change to treatments.
In recent years, a growing number of researches regarding CF related psychological diseases were carried out. - replace with studies
The International Depression/Anxiety Study (TIDES) revealed a high prevalence of scores associated to psychological diseases in adult and adolescents with CF - statement is somewhat confusing. needs further explanation or rewording.
Round 2
Reviewer 1 Report
The authors have made an effort to address some of the comments and recommendations outlined in the earlier review. However, it should be noted that it is very difficult to identify and find specific changes when large sections of text are highlighted, no tracked changes provided and no clear response document to the reviewer submitted alongside the revised manuscript.